# Interdisciplinary collaboration from diverse science teams can produce significant outcomes

**Alison Specht**[1☯], **Kevin Crowston**[2☯]*

**1** Terrestrial Ecosystem Research Network, The University of Queensland, Brisbane, Australia, **2** School of Information Studies, Syracuse University, Syracuse, NY, United States of America

☯ These authors contributed equally to this work.

* crowston@g.syr.edu

**Data Availability Statement:** The data availability statement is aligned with the ethics requirements for the study. There were several limitations imposed by the Institutional Review Board of Syracuse University and accepted by the University

## Abstract

Scientific teams are increasingly diverse in discipline, international scope and demographics. Diversity has been found to be a driver of innovation but also can be a source of interpersonal friction. Drawing on a mixed-method study of 22 scientific working groups, this paper presents evidence that team diversity has a positive impact on scientific output (i.e., the number of journal papers and citations) through the mediation of the interdisciplinarity of the collaborative process, as evidenced by publishing in and citing more diverse sources. Ironically these factors also seem to be related to lower team member satisfaction and perceived effectiveness, countered by the gender balance of the team. Qualitative data suggests additional factors that facilitate collaboration, such as trust and leadership. Our findings have implications for team design and management, as team diversity seems beneficial, but the process of integration can be difficult and needs management to lead to a productive and innovative process.

## Introduction

Complex, global research challenges increasingly require the formation of teams that bring together individuals with diverse skills and perspectives who can work across disciplinary, organisational and national divides [1–5]. One form for these teams is a working group, a group of researchers from different institutions who are brought together and supported to work over a period of a few years on an (often self-identified) compelling research problem. The outcome of the team work may be to inform policy development, to produce new knowledge and understanding, or the creation of e-infrastructure platforms to support continued advances in science [6]. Such teams should not be mistaken for an established research team given a specific task within their work schedule as part of their employee duties.

Working groups can be assembled in a way that creates several kinds of group diversity. A kind of diversity that is increasingly important to science is disciplinary diversity [7]. In recent years interdisciplinary research has rapidly become the norm [8, 9], that is, research involving scholars from different disciplines collaborating to develop terminology, research approaches, methodologies, or theories that are integrated across multiple disciplines to address broader

of Queensland as suitable ethical conduct for the work proposed (Syracuse University #13-202, 16-203 and 18-230). These were as follows: "Personal information that is collected will be used solely to enable network analysis of members within a working group and will not be used for any other purpose. Results of the research may at some future time be published. Although your responses may be identifiable to the researchers, responses will be kept confidential and no individual responses will be reported; only summarized findings will be reported." and "Your identity will be held in confidence as an invitee to the survey associated with the relevant synthesis centre and group. Your identity will not be published." This being so, we have anonymised all respondent identities, including identifiable links to their organisations and disciplines, and to the specific publications by each group (which would allow identification of the authors and hence the groups being studied). We have not published here or in the data repository for the paper the full demographic profiles of members. With that consideration, the data are available as follows in the Environmental Data Initiative [83]. The data themselves will remain anonymised.

**Funding:** The authors received no specific funding for this work.

**Competing interests:** The authors have declared that no competing interests exist.

problems than a single disciplinary approach can address [8]. There is an accepted understanding that interdisciplinary research will foster important ideas beyond the boundaries of a single discipline, even creating new disciplines [7, 10–12]. Further, there is some evidence that increasing the interdisciplinarity of the research team will increase the originality and creativity of the outcomes [7, 13, 14]. Certainly an interdisciplinary team is thought likely to be more creative and produce more novel results than conventional research teams due to the variety of disciplines being blended [15–20].

In addition to disciplinary variety, other kinds of group diversity have impacts on group process and outcomes. Combinations of participants from different kinds of organisations can result in technology fusion, creating new opportunities for each participant [21, 22]. International collaboration in the environmental sciences has often been required due to the nation-boundary-blindness of organisms and processes. In addition, there are often compositional requirements, such as representative membership, as is the case in most European Union-funded projects, or to comply with societal expectations, such as gender and race representation.

While group diversity can provide the pathway for innovative insights, diverse teams face challenges to collaboration, for example friction, lack of communication and break points, such as those due to differing cultural norms around gender, power, and the preferred level of individual behaviour versus collectivism [23]. Group members in new combinations may find more questions than answers for a particular problem, defeating their purpose. Effective collaboration across disciplinary or national boundaries does not result from simply 'putting people together in a room and shutting the door'. For diverse groups of people to make sense of the vast range of data and information increasingly available and generate an outcome that is innovative and creative, thoughtful group construction and support is required.

The goal of this paper is to identify key aspects of group composition conducive to productive work around multi-disciplinary, multi-layered problems. Is team diversity positively related to outcomes that combine the team's diverse backgrounds and is the productivity of such a team enhanced by the level of diversity within it? If so, what aspects of diversity might be most important? Task-related diversity has been found to be positively related to performance, but bio-demographic diversity has not [24]. Does having a certain gender balance help? Does diversity itself create difficulties of communication, and impede cross-fertilisation across sectors and hence output? How might this work if more than one factor occurs at once?

## Theory development

As a basis for our analysis, we develop a model of the factors that may predict the success of interdisciplinary science teams through providing the basis for the development of group collective intelligence [20, 25, 26] and the intellectual fusion or, as some would express it, integration, that is desirable to generate transformative outcomes [27, 28]. We are guided by the Input-Process-Output group framework (Fig 1). A version of this input-process-output model was used by Stokols et al. [29] in their examination of 'transdisciplinary scientific collaboration'. In this section, we explain the model and the specific variables we chose for each aspect of the model.

In this model, underline{inputs} are the attributes that the team members bring to the teamwork. In our setting, we are focusing on the members themselves and their demographic backgrounds. A working group is composed in response to a problem: people are deliberately chosen with relevant skills and expertise to contribute to a solution, while keeping in mind other factors, such as organisation, country, and life situation. These choices create groups with different levels of diversity.

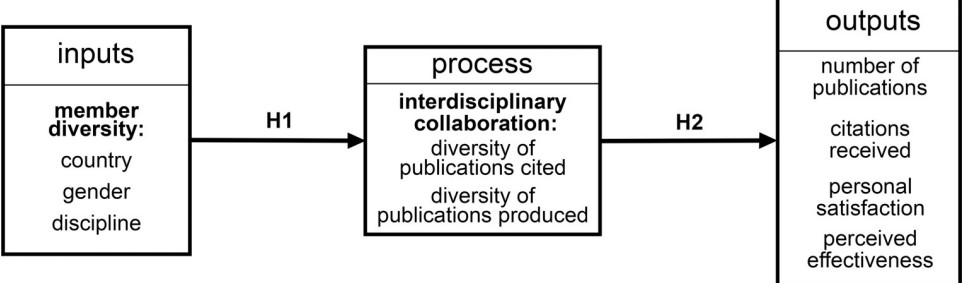

**Fig 1. Research model.** Inputs, mediators (process and emergent states) and outputs. Hypothesis numbers are shown.

Outputs of a group are the measurable consequences of team function. For research groups the desired outcomes are new knowledge and understanding expressed usually through refereed publications, deposited data in open data repositories, and new, long-term collaborations across disciplinary, organisation and geographical boundaries. Lynch et al. [30], show a range of types of valued outputs from synthesis centers. As noted also by Hackman [31, p. 128], team outputs can extend beyond specific products to include participant satisfaction with their work and their perceptions of the effectiveness of the team.

Between inputs and outputs is the process that mediates the influence of inputs on outputs. The measurable and intended process in our study is that of collaboration across disciplinary boundaries. Interdisciplinary or trans-disciplinary group membership is often used to support cross-fertilisation to "co-design research, co-produce solution-oriented knowledge, and reintegrate the knowledge into strategies for problem-solving and the development of new scientific insights" [32]. The details of these interactions are subtle and difficult to capture in a comparable way across teams. Research has therefore usually adopted proxy measures. In this paper we assess interdisciplinary research output by the diversity of the authors and of the references they cite [33, 34].

## Hypothesis development

Much of the research that has addressed the question of the effect of team composition on outputs has examined diversity directly without postulating an intervening process. Publication output, for example, has been positively related not only to disciplinary diversity but also low seniority of teams [35]. Others have found a positive effect of diversity of career stages and gender on publication output, but a negative effect of educational (disciplinary) diversity [36]. Gender heterogeneity in a group has been shown to enhance citation rates [37] and link positively with team performance [38]. The novelty of our approach is that rather than hypothesizing direct impacts of diversity on outputs, we expect outcomes to be moderated by the nature of the research process. We consider that different kinds of diversity may contribute in different ways and so develop a set of hypotheses for each of these relationships.

**Inputs to process.** We first consider how group diversity (the input) may have an impact on the group process, specifically the interdisciplinarity of the collaboration (the process).

Although men and women share common predictors for collaboration, women have been found to tend to engage in more interdisciplinary research collaborations [39], to have more collaborators than men [40] and they have characteristically more social sensitivity [25, 31]. Indeed, in studies of small groups (7–10 members) the collective intelligence of the group has been positively correlated with the percentage of women in a group [25], as has group effectiveness and innovation, particularly when task intensity is high [40, 41]. This result has been

**Table 1. Hypotheses posed in this paper.**

| Hypothesis | Proposed relationship |
|---|---|
| H1 | There is a **positive** relationship between interdisciplinary collaboration and: |
| | (a) proportion of female members |
| | (b) the interdisciplinarity of the group |
| | (c) There is a **negative** relationship between interdisciplinary collaboration and the diversity of international membership |
| H2 | (a) There is a positive correlation between interdisciplinary collaboration in a group and number of publications. |
| | (b) There is an inverted-U relationship between interdisciplinary collaboration in a group and impact as measured by the median number of citations. |
| | (c) There is a positive correlation between interdisciplinary collaboration in a group and personal satisfaction |
| | (d) There is a positive correlation between interdisciplinary collaboration in a group and perceived effectiveness |

explained by improvements in the emotional intelligence of the group and a reduction in conflicts with the addition of a component of women [38, 42]. We therefore hypothesise that interdisciplinary collaboration will be enhanced by the proportion of female participants (Table 1, H1(a)).

Without a range of disciplines present in a group—as assessed by a classification of the group members into their 'home' disciplines [2]—the meshing of disciplines could not occur. It has been found that the diversity of citations tends to grow with the diversity of disciplines of the authors [33]. We therefore hypothesise that interdisciplinary collaboration will be enhanced by the interdisciplinarity of the group (Table 1, H1(b)).

Finally, intra-country (regional, state) or international membership may also be required for project performance, depending on the problem being considered or the funding arrangements. In environmental and biodiversity science and management, global participation in some shape or form is essential not the least because organisms do not abide by human political boundaries, but also (i) to get a comprehensive grasp on a problem, (ii) to acquire the requisite data, and (iii) to access appropriate expertise and data. Indeed, international collaboration has become the norm rather than the exception in modern science [3, 34]. Group membership may be mandated or perceived as desirable for funding reasons (e.g. funding from the European Union or the Belmont Forum) along with a perception that the results will be enhanced by such inclusivity.

While there may be a clear need or desirability for international collaboration, there are clear challenges to successful collaboration. The physical dispersion of group members creates logistic problems due to travel approvals and cost, and if collaboration is to be remote, timezone challenges can be prohibitive [1, 16]. Participants from different countries with different languages and ways of working can have communication and cultural challenges which take time to overcome, reducing the productivity of the group. This can result in less novel outcomes, and high transaction costs and communication barriers; for instance, data suggest that the more nations involved in a project, the fewer publications and the more conventional the research [3]. We therefore hypothesise that interdisciplinary collaboration will be negatively affected by the diversity of international membership (Table 1, H1(c)).

**Process to outputs.** We next consider how the process affects the outputs, in the first instance, the number of publications produced. Collaboration in general has been positively related to higher output [43, 44]. Researchers engaging in interdisciplinary research have been found to be less productive but have higher impact [45]. In contrast, authors who publish at

moderate levels in disciplinarily diverse journals and with a moderate level of collaborator diversity have been found to publish more, while a moderate level of collaborator diversity is beneficial for authors who are more focussed on output [46]. We draw on these two streams of literature to propose that there is a positive correlation between interdisciplinary collaboration in a group and the number of publications produced (Table 1, H2(a)).

Our second output measure is the impact of the work as indicated by citation rates. Several articles have examined the effect of team diversity on the citations received (e.g., [47–49]). We conceptualize these factors as having an impact via their effect on the interdisciplinarity of the team process. The effects of interdisciplinarity have mostly been studied at the level of individual articles rather than teams, generally finding a positive impact of citing more diversely (e.g., [50–52]). In contrast, at the individual researcher and team level, the effect has been found to be curvilinear, with an inverted U-shape, meaning that impact is highest for intermediate levels of interdisciplinary [53, 54]. The effect has been attributed to the costs of learning to work across disciplines, where too much interdisciplinarity can become a distraction (Table 1, H2(b)).

In the previous two hypotheses, we have argued that teams with an interdisciplinary process will be more successful in publishing. We expect this success to be reflected in the evaluations of the team members of group function. For example, that there will be a positive relationship between the interdisciplinarity of the collaboration in a group and member satisfaction with the group (Table 1, H2(c)), and the perceived effectiveness of the group (Table 1, H2(d)).

## Methods

The project follows a mixed methods approach using triangulation of quantitative results with qualitative survey information [55]. The quantitative study examines team demographics, processes and outcomes while the qualitative component examines the opinions proffered by team members about the reasons for group success or otherwise.

Before commencing the detailed description of the data acquisition for the reported project, the limitations related to the ethics statements made to the participants and accepted by the Institutional Review Board of the corresponding author's university and complying sufficiently with the Australian ethics standards were as follows:

"Personal information that is collected will be used solely to enable network analysis of members within a working group and will not be used for any other purpose. Results of the research may at some future time be published. Although your responses may be identifiable to the researchers, responses will be kept confidential and no individual responses will be reported; only summarized findings will be reported." and "Your identity will be held in confidence as an invitee to the survey associated with the relevant synthesis centre and group. Your identity will not be published." This being so, we have anonymised all respondent identities, including identifiable links to their organisations and disciplines, and to the specific publications by each group (which would allow identification of the authors and hence the groups being studied). We have not published here or in the data repository for the paper the full demographic profiles of members.

## Research setting

We test these hypotheses through studies of a selection of research working groups designed to have some degree of diversity, with members with a variety of disciplines, skillsets and origins. The research working groups we are focussing on are a facilitated version of the 'self-assembly' to which Twyman and Contractor [56] refer and which are well-discussed in the synthesis center literature [15, 30, 49, 57]. These groups gather, with logistic support, to analyse a specific

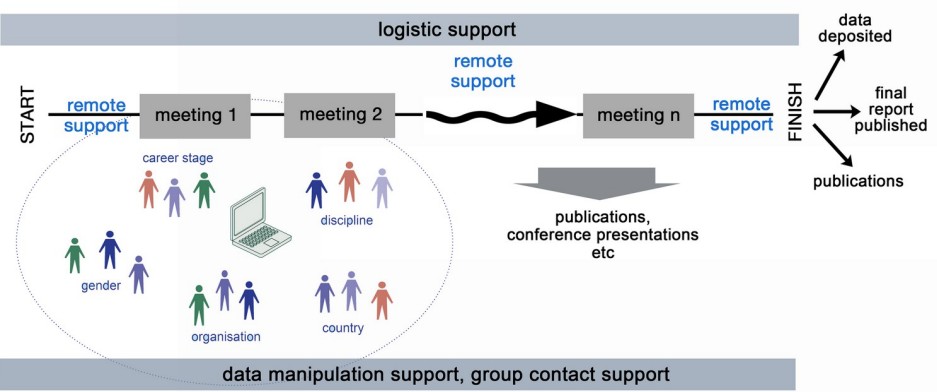

**Fig 2. Working group workflow.** A depiction of the way the working groups work together in our scenarios, where groups of diverse people meet together intermittently over a period of time in a supported environment to examine difficult, multi-disciplinary, problems. The meetings might each last up to a week, and are often at six month to yearly intervals.

complex problem, drawing on members from many institutions and countries for intermittent but concentrated periods of time apart from their normal workplace commitments. The staccato nature of the short but concentrated time the groups spend with each other in these 'hot moments' is often termed 'island time' (Eric Garnier pers. comm. in [15]) and has been found to be particularly conducive to creativity [26, 57]. The members of such groups are largely 'volunteers', driven by a common interest in the problem being addressed. They are supported to a limited extent in their endeavours, but not salaried to solve the problem at hand (Fig 2).

The groups we studied were drawn from a large cyberinfrastructure project (DataONE) and from two synthesis centers in the biodiversity and environmental fields. The DataONE project adopted the synthesis working group approach in its participatory engagement model, enabling it to tap into expertise in the wider community as it constructed its e-infrastructure [58].

Synthesis centers are purpose-built organisations designed to enable diverse groups of people (working groups) to synthesise disparate data and information on a particular topic to produce new understanding [17, 49, 59, 60]. They have been likened to the business 'incubator' [61, 62]. Synthesis centers support the use of existing information and apply, to greater or lesser extent, many of the elements of team science, open science and data-intensive science [5, 15, 30, 62]. They have been highly successful in facilitating collaborations and are highly productive [15, 17, 49, 60, 63].

The composition of synthesis center groups is largely determined by group leaders (Principal Investigators) within criteria set by the centers, such as a degree of inter-(or less often, trans-) disciplinarity, multi-national and multi-sectoral teams, career and gender balance ([51] and as described on the center web sites accessible through www.synthesis-consortium.org). This approach means that a large proportion of the group starts with agreement on a common goal. The working groups in our study were supported for a maximum of four years with multiple meetings over that time. For the synthesis centers and for DataONE, good outcomes are critical to continued funding and the organisations post the products of the groups they sponsor on their web sites for public access.

It is relevant to note that the work of these groups is collegiate, as far as communication and the development of trust will allow. Decisions are made as a group, for example, once the groups have examined the data and information to hand, they will determine their work plan,

identifying the products (usually articles but some code and conference presentations) that would seem most achievable and valuable, and then separate the tasks among the group members according to the skill sets needed. Occasionally needed skills were not available within the group and additional experts were brought in to complete the specific task. As one survey respondent put it, "The group was very effective at dividing tasks, assigning roles, and then getting their parts done. Everyone wanted to contribute." (C-1, R2). If a group member did not pull their weight they could be isolated. The group leader(s) keep track of these tasks and all group members will revisit their various tasks at the start of each meeting.

By confining our study to this cohort we were able to compare across three similarly-aged organisations all providing similar logistic and infrastructure support (an important factor influencing creativity of output according to [14]). The selection of the specific groups for study within each Center or project were determined by (a) advice of the director of the Center (JWP), (b) maturity of the group, and (c) response level to the qualitative survey (described below).

It should be noted that although our ambitions were to include another center and hence suite of groups in our study, the response from the center approached, although positive, was tardy (one year after initial enquiry) and did not fit our time-frame, especially given we already had results from 2014. The size of the sample we analyze in this paper restricts the analysis possibilities.

## Data acquisition

We collected three types of data for the model:

1. demographic profile of group members (<u>inputs</u>: Fig 1)

2. collation and analysis of articles used and produced (<u>process and outputs</u>: Fig 1)

3. group member perceptions (<u>outputs</u>: Fig 1)

For items 1 and 2 the data were available through the group websites. Item 3 was accomplished by a survey sent to the members as published on the group websites.

**Working group demographics.** The demographic profile of all members of each selected Working Group was collated. This included their country of origin and their gender (binary only; no members with non-binary identities were identified).

The primary scientific discipline of each group member was defined using their own statements to the contributing organisations, and where not available, their self-stated fields on sources such as ResearchGate and Google Scholar, and their pages on their organisation's web sites. An initial list of forty-six primary disciplines was created using the Australian and New Zealand Fields of Research [64] to derive a smaller controlled vocabulary of 22 categories. This reduced vocabulary included biology and applied biology, chemistry, climate science, communication, computing, data science, earth sciences, ecology, education, engineering, environmental science, evolution studies, freshwater (and marine) studies, geography, health, hydrology, library and information studies, modelling, policy, sociology, and statistics. As mentioned, for confidentiality reasons these discipline fields are not listed to enable linkage with the participant or the group to which the member belongs.

**Collation of articles used and produced.** A key construct in our model is the interdisciplinarity of the group process. As we were unable to follow each group individually over the several years they were collaborating, we sought a measure that could be applied retrospectively. Interdisciplinarity can be assessed in many ways [5], of which bibliometric measures are the most developed and have valuable antecedent analyses [e.g. 65–69]. We used journals as an

indication of discipline. Journals are responsive to the communities they serve (they have to have an 'audience' to survive), and several disciplinary themes or 'keywords' are used to describe the journal field, thus encouraging relevant articles and providing appropriately qualified editors and reviewers to assess the submissions. In this way the selection of journals in which to publish is tailored to the subject matter. Of course, some journals are quite general and welcome a wide range of articles, which clouds the specificity of the domain description, although in our case these occur rarely in our study due to their high impact (e.g., Nature and Science). Other measures such as data deposition, grant success and conference presentations are also useful indicators of productivity as well as measurable for disciplinary variety, but they were inconsistently measured across our study groups, so we have confined ourselves to refereed journal articles produced by the groups.

We based our analysis on the groups' publications. We obtained copies of all papers published by the groups up until August 2020, the vast majority of which were open access. The number of citations to each publication was acquired from CrossREF on 19 August 2020. Not all articles produced by the groups were in journals tracked fully by CrossREF, as CrossREF mainly scans English-language journals, but this is a relatively minor impediment to our analysis as English is the norm for the fields we are studying.

To assess the interdisciplinarity of the working groups' collaborations, we analysed the literature on which the group drew to produce their published articles, the 'inspiration' measure of [9] and used by several authors in whole or in part as a measure of interdisciplinarity [33, 70–72]. Specifically, we examined the disciplinarity diversity of the cited journals [76, 73]. We selected this measure based on the premise that groups working in an interdisciplinary way bring together diverse knowledge [2, 74, 75], which will be reflected in the diversity of the literature cited. To consider the inter-or trans-disciplinary nature of the outputs from a group rather than just their productivity, we also examined the degree to which the outputs appear in journals in a diversity of disciplines [49, 76].

To obtain these measures, we first extracted which journals were cited in the published papers, as well as the journals in which the groups published their own articles. A total of 1007 journals were identified in this process. Each journal was classified according to disciplinary category(ies) using SCOPUS, SCIMAGO and journal-stated discipline fields if not listed in those databases, being generous rather than reductionist in allocation. In this manner 128 specific journal disciplines were identified across all working groups, from agriculture to ecology to parasitology and water science. Guided by the Australia and New Zealand Fields of Research categorisation (ANZFoR [64]), these specific disciplinary categories were then clustered under 22 ANZFoR disciplinary divisions, with two non-ANZFoR categories created due to no suitable matches in the ANZFoR Divisions. These discipline categories for journals, although they are responsive to the communities they serve, are admittedly relatively general. The disciplinary divisions used were: agriculture, veterinary and food sciences, biological sciences, biomedical and clinical sciences, built environment and design, commerce, management and tourism services, earth sciences, economics, education, engineering, environmental sciences, health sciences, history, heritage and archaeology, human society, information and computing sciences, language, communication and culture, law and legal studies, mathematical sciences, philosophy and religious studies, physical sciences, and psychology. The two additional divisions were 'bio water science' which covered all freshwater and marine biological sciences, and 'general science' which enabled the classification of journals like Science, and Nature that cover multiple disciplines.

**Group member perceptions.** A third source of information came from a survey of Working Group members who were asked via electronic survey about their perceptions of group performance (satisfaction and perceived effectiveness on a 1–5 point scale, from not at all

effective/satisfied to very effective/satisfied, with an invitation to make comments explaining their choice). The initial intent of these questions was to combine them with productivity as a multi-item measure of group success [31], but as the correlations were not high, we instead analyzed them individually. We also invited open-ended comments from subjects about the 'primary factors that they felt contributed to their working group's effectiveness or lack thereof', for a total of five questions. These questions were part of a longer survey including questions not related to interdisciplinarity that we do not analyse in this paper. As there were many other components in the survey, we considered that having too many items for these scales would negatively impact the return rate. Links to the entire questionnaire were emailed first to the DataONE email distribution list during an All Hands meeting in October 2014 (including to members who were not in attendance at the meeting) at the end of Phase 1 of DataONE. The same survey was subsequently emailed to Working Group members of the two synthesis centers in late 2018. Only groups that had finished or were close to finishing their work were invited to participate and there was little change in group composition at the times of the surveys (2014 and 2018) in relation to when the groups started (and finished). In all cases, 2 weeks was given for return, with an extension of another week. We only included groups in our analysis from which we received survey responses from at least 20% of members.

## Analysis

**Operationalization of concepts.**   In this section we explain how we used the collected data to measure our research constructs. First, we computed measures of group diversity along the different demographics. For each group, we started with counts of members in different demographic categories (e.g., number of members from different countries). For country and discipline, we used the category counts to compute a measure of diversity (entropy) using the Shannon index [49, 77–79]:

$$\text{Diversity or entropy}: \ H = -\Sigma(p_i \ln p_i) \tag{Eq1}$$

where $p_i$ is the proportion of members in group i.

Since the data collected on gender were binary, we simply used the proportion of female members in the group. In our dataset the range of proportion female was 4–67%, meaning we had some nearly all-male groups but no all-female groups.

To assess the extent of the interdisciplinarity of the collaboration process, we measured the diversity of the disciplines of the journals in which the groups published and those they cited in their publications (computed using Eq 1 above). For this purpose, we combined the count of journals cited per discipline across all the working group's publications. Diversity can, of course, be assessed in different ways, such as variety, balance and disparity [79]. In a similar manner to assessment of the variety of demographics, we used Shannon diversity as our measure for disciplinary diversity. Although this measures variety it is also affected by balance (lower balance leads to lower diversity). We also note that some studies of interdisciplinarity have assessed not just the variety of disciplines cited but also the atypicality of the combination (e.g., [80]), however, since our classification of journals is based on a different system than in Uzzi [80], we do not have data on typicality.

We measured the output of the group in four ways, first by the number of publications and the impact of the group's work by computing the median number of citations the various publications attracted. We assessed satisfaction with group performance and perceived group effectiveness at an individual level through a survey of members. We used the mean of the individual scores as the group measure.

In addition to the variables in the model, we included two control variables: (i) the age of the group, with the assumption that groups that have been around longer have had more of an opportunity to publish and to have an impact (as measured by citations), and (ii) the Center (a three-level factor variable), to control for different expectations around publishing in the three settings.

**Hypothesis testing.**   Hypotheses were tested using regression on the data emerging from the demographics and diversity measures. Data other than counts were standardized before regression. One problem arose in carrying out the regressions: as we had only 22 groups, using too many variables in the regressions led to overfitting. Unfortunately, the small number of data points also meant that we could not test the hypotheses simultaneously, e.g., with a structural equation model.

The problem of potential overfitting arose for the analyses of the demographic input variables. A regression that included all the demographic variables achieved nearly perfect $R^2$, an indication that the model was overfit. To avoid this problem, we used a stepwise regression approach, adding variables that were most related to the outcome, but stopping with a small number of variables. We explored reducing the dimensionality of the input variables through factor analysis but did not find a satisfactory solution with a smaller number of factors. We also explored more modern techniques for variable selection such as lasso but did not have enough data to use them.

Where we hypothesised curvilinear relationships, we entered variables both as a linear and a squared term using the R poly function, which computes orthogonal polynomials to avoid multicollinearity. As the use of this function complicates interpretation of the regression coefficients, we present non-linear relationships graphically.

**Qualitative analysis.**   The open-ended responses were subject to thematic analysis related to the research question ("In your opinion, what are the primary factors that contributed to your working group's effectiveness or lack thereof?") using an inductive semantic approach [81]. The themes thus identified alongside their associated (anonymised) comments were sent to the groups for validation of our interpretation and modified if required. In this process additional insight was often obtained.

## Results

### Group analysis

**Demographic data.**   Demographic profiles were obtained for the groups that responded to the on-line surveys in sufficient numbers (>20% responses). The resulting population of 389 people came from 28 countries, the majority from the USA (62%), and the next highest from France (13%), the home of the third organisation in the study. The UK and Canada were also relatively well represented. The total population was predominantly male (68% male and 32% female) and 51% of members were from universities, 19% were from government organisations, and 15% from research organisations.

Ecology was by far the most common primary discipline type (23%), with an equal proportion of people in the computing, data science, statistics and modelling areas. These distributions reflect the focus of the sponsoring organisations and the emphasis of the working groups on working with data. Freshwater biology and ecology were clustered together into freshwater science, which contributed 4.6% of the total.

Group size was variable, ranging from 11 members up to a maximum of 28 (S1 Appendix). The demographics differed between groups, with some groups very international, and others exclusively from one country. The proportion of females in the groups ranged from 0% to 67%, an average of 33% ± 4%. While there are clear differences among groups, overall the mix

of respondents seems representative of scientists who participate in working groups [18, 28, 42] and was consistent with the first author's personal experience coordinating working groups. Groups that had at the time of the survey tended not to respond to the survey (nor to have output to analyse), and so are not included in the study. Nevertheless, we have a range of productivity and impact in the groups included.

Meeting attendance data (not that that is the only measure of fidelity to the group) was not available consistently for all groups, but for the data that were available, fidelity to the group was between 60% and 80% over 2–10 meetings. Fidelity was always greater than 80% for groups that had only two meetings.

**Publication analysis.** One hundred and fifty-seven journal articles were collectively produced by the groups at the time of our study (August 2020), and 6,749 articles were cited in these articles. The articles appeared in 83 different journals in 22 ANZFoR disciplinary divisions. The articles cited (the 'inspiration' of Gates et al. [9]) were drawn from journals in 18 ANZFoR disciplinary divisions.

Group A-1 had the lowest publication diversity of all the groups at 0.693 (S2 Appendix) with only two publications at the time of the study, one in the ANZFoR Division biological sciences and one in environmental sciences. The citation diversity was similarly low (0.287, S2 Appendix). The articles cited across these two papers were from forty-six journals, with thirty-two journals cited in one paper and seventeen cited in the other. The vast majority of the citations (91% and 93%) were to publications in the ANZFoR Division biological sciences.

In contrast, B-7, with eighteen publications at the time of the study, had the highest publication diversity at 1.874 (S2 Appendix) with publications across seven ANZFoR divisions (agriculture, veterinary and food sciences, biological sciences, earth sciences, environmental sciences, general science, human society, and information and computing sciences). To produce these articles, the group drew on papers from seventeen ANZFoR Divisions (agriculture, veterinary and food sciences, bio water science, biological sciences, biomedical and clinical sciences, built environment and design, commerce, management and tourism services, earth sciences, engineering, environmental sciences, general science, history, heritage and archaeology, human society, information and computing sciences, language, communication and culture, philosophy and religious studies, physical sciences, and psychology). The citation diversity (and its distribution amongst the papers), however, was not the highest of the groups at 1.301 (S2 Appendix). Group C-4 had the highest citation diversity with an index of 1.656 drawing on articles from ten ANZFoR Divisions (agriculture, veterinary and food sciences, bio water science, biological sciences, earth sciences, engineering, environmental sciences, general science, information and computing sciences, mathematical sciences, and psychology) across their three articles.

Citations per article ranged from a high of ninety-six different journals for paper C of group A-6, to no, or one, citation per article (e.g. B-6 paper C and B-3 paper F), while many groups cited forty or fifty articles per paper (e.g. C-1 paper A, A-9 paper F, B-1 paper B).

The average number of authors per article was 8.75 ± 0.51. This number alone is greater than the average number of authors for all journal discipline fields found in past studies of scientific collaboration. For example, a maximum of 5.9 authors per article for the environmental sciences was reported in Patience et al. [82] and 5.19 for synthesis centers by Hackett et al. [49]). This evidence suggests the working groups in our study were highly collaborative. The average number of authors per article was highest in group A-4 closely followed by C-1 and A-9, while the minimum average number of authors was 1.7 for group B-6 (S2 Appendix).

The bibliographic data are available in the repository of the Environmental Data Initiative [83].

**Qualitative feedback.**   Response rates to the surveys per group ranged from 56% to a minimum of 20% (our cut-off for including the group in the study). Of the ninety-two respondents who provided feedback to the ranked questions about group effectiveness and satisfaction, only forty-eight provided open-ended responses about effectiveness, and twenty-two, about satisfaction. One hundred and twelve responses were received to the general request for comment across all three survey groups. Detailed commentary on the open-ended responses is included in the qualitative analysis.

Respondents' perception of their group's effectiveness and satisfaction was positive (all measures were above 2.5 on a 5-point scale, S3 Appendix)). However, some groups (A-7 and A-8, for example, and B-5) were as low as 3 on the 5-point scale, while others (A-4, B-6 and C-2) thought their groups were particularly effective and were correspondingly satisfied with the group's function. There was not, however, a linear relationship between the two metrics, so we treated them as separate entities.

## Hypothesis testing

In this section, we describe the results of testing the proposed hypotheses.

**Prediction of interdisciplinary collaboration (H1).**   To assess the level of interdisciplinary collaboration in the groups we used the diversity of the disciplinary fields of the references cited by the groups, and the diversity of journals chosen by the groups for their publications. We used stepwise regression to identify influential variables for each dependent variable separately.

The disciplinary diversity of the journals cited by the groups in the production of their articles (cited publication diversity) was positively related to the disciplinary diversity and the proportion of women in the group ($p < 0.01$; Table 2). The diversity of the journals in which the groups published was negatively related to the county diversity of the group ($p < 0.05$; Table 2) and positively related to discipline diversity ($p < 0.05$), but other factors were not selected by the regression.

**Prediction of output (H2).**   As the outcome variable was a count, a Poisson regression was used to assess effectors of the number of publications produced. On the other hand, the median number of citations received was over-dispersed (the standard deviation was greater than the mean), with a few clear outliers, so a negative binomial regression was used instead.

The disciplinary diversity of the journals in which the groups published predicted the number of publications ($p < 0.01$, Table 3). The disciplinary diversity of the articles cited in

**Table 2. The effect of input variables on citation and publication diversity.**

|  | Diversity of publications cited | Publication diversity |
|---|---|---|
| Intercept | 0.000 (0.122) | 0.000 (0.165) |
| Country diversity |  | −0.403* (0.175) |
| Proportion of female members | 0.644*** (0.130 |  |
| Discipline diversity | 0.391** (0.130) | 0.445* (0.175) |
| Number of observations | 22 | 22 |
| adjusted $R^2$ | 0.669 | 0.401 |

Mean, standard error (in parentheses) and significance.

+p < 0.1

* p < 0.05

** p < 0.01

*** p < 0.001

**Table 3. Factors predicting output variables.**

|  | Number of publications | Median number of cites to |
|---|---|---|
| Intercept | 2.105*** (0.160) | 2.351*** (0.241) |
| Project age | 0.818*** (0.156) | 0.091 (0.220) |
| A groups | – | – |
| B groups | −1.381*** (0.342) | 1.854*** (0.473) |
| C groups | 0.329 (0.287) | 0.623 (0.451) |
| Publication discipline diversity | 0.257** (0.099) | 0.608 (0.767) |
| Publication discipline diversity squared |  | −2.434*** (0.636) |
| Cited discipline diversity | 0.372* (0.152) | −1.788* (0.840) |
| Cited discipline diversity squared |  | −1.627+ (0.831) |
| Number of observations | 22 | 22 |
| Nagelkerke's pseudo $R^2$ | 0.888 | 0.940 |

Regression weight, standard error (in parentheses) and significance of each variable is shown. "A" groups were used as the baseline for the group factor.

+ $p < 0.1$

* $p < 0.05$

** $p < 0.01$

*** $p < 0.001$

producing those publications, our measure of interdisciplinary collaboration, also positively predicted the number of publications ($p < 0.05$, Table 3). As hypothesised, the median number of citations received had inverted-U relationships with both measures of interdisciplinary collaboration, as shown in Fig 3. Among the controls, there was a strong linear positive relationship between group age and the number of papers produced ($p < 0.001$; Table 3) and differences were noted among the centers. Specifically, the B groups had significantly fewer publications but more citations than the A groups, while the C groups were not significantly different (note that the choice of the A groups as the baseline to which B and C are compared is arbitrary).

**Perception of satisfaction and effectiveness.** A model predicting self-reported satisfaction and perceived group effectiveness based only on process and emergent state variables (the

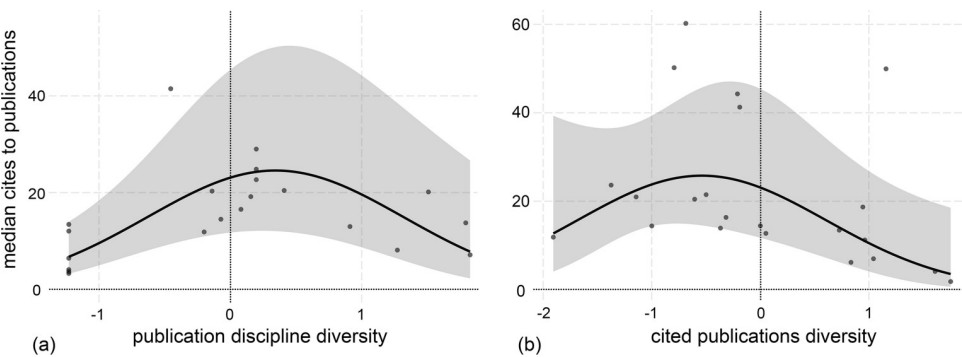

**Fig 3. Influences on citations received.** The median number of citations received by a group's publications against (a) publication discipline and (b) the diversity of publications cited. One group is an outlier to the plot in 4(a) and is not visible in the plot. The 95% confidence envelope around the trend is shown. '0' on the axes is the grand mean for all groups for that variable, and the intervals shown are one standard deviation from the mean. The data points show the partial residuals for the groups, i.e., the residual after controlling for the other variables in the regression.

**Table 4. Predictors of satisfaction and perceived group effectiveness.**

|  | Perceived satisfaction | Perceived effectiveness |
|---|---|---|
| Intercept | 0.000 (0.164) | 0.017 (0.159) |
| Project age | −0.572* (0.192) |  |
| ln (number of publications) | 0.462* (0.209) | 0.238 (0.168) |
| Publication discipline diversity | −0.404 (0.255) | 0.376 (0.213) |
| Cited discipline diversity | −0.817* (0.313) | −0.512* (0.199) |
| Country diversity | −0.626* (0.242) | −0.260 (0.176) |
| Proportion female | 0.624* (0.289) |  |
| Disciplinary diversity | 0.316 (0.247) |  |
| Number of Observations | 22 | 22 |
| Adjusted $R^2$ | 0.422 | 0.169 |

Mean, standard error (in parentheses) and significance of each variable shown.

+ $p < 0.1$

*$p < 0.05$

**$p < 0.01$

***$p < 0.001$

hypothesised model) did not reveal any significant predictors. We therefore considered whether these outcomes might be affected also by group composition. To test this *post-hoc* hypothesis, we used stepwise regression to identify independent variables. We also added the number of publications (log transformed because of skew) as a predictor.

Satisfaction with the group was negatively related to the age of the group ($p < 0.05$; Table 4) and, as might be expected, positively related to the number of publications the group produced ($p < 0.05$; Table 4). Satisfaction was negatively related to the diversity of publications cited in the production of those papers ($p < 0.05$; Table 4), negatively related to the diversity of countries of the group members ($p < 0.05$; Table 4), and positively related to the proportion of female members in the group ($p < 0.05$; Table 4).

Predictors of perceptions of the effectiveness of the group were more limited, namely that the more diverse the publications sourced by the group the less effective the group was perceived to be ($p < 0.001$; Table 4), the higher disciplinary diversity of the group, the more effective respondents felt the group was ($p < 0.05$; Table 4), and there was a moderate positive influence of the proportion of female members ($p<0.05$; Table 4).

**Summary.** All hypotheses were supported except for Hypotheses 2(c) and 2(d) (Table 5). Country of origin negatively affected the process of publishing diversely and personal satisfaction with the group (Fig 4). Citing a diverse range of articles in producing output was correlated with lower satisfaction and perceived effectiveness, despite not actually affecting measurable output, which had a strong positive effect on personal satisfaction with the group (Fig 4). The proportion of female members had a positive effect on the diversity of articles cited and on personal satisfaction with the group (Fig 4). The diversity of disciplines represented in the groups had a strong positive effect on both metrics used for interdisciplinary collaboration and had a less positive effect on the perceived effectiveness of the group (Fig 4).

## Qualitative analysis

The outcomes from the open-ended feedback to the survey of member's satisfaction with and perceptions of the function of the group did not always compartmentalise neatly under our hypotheses nor did they directly complement the quantitative results. They added, however,

**Table 5. Summary of the outcome of the hypothesis tests.**

| Hypothesis | Proposed relationship | Comment |
|---|---|---|
| H1 | There is a **positive** relationship between interdisciplinary collaboration and: | |
| | (a) measures of gender diversity | The more female members in the group, the greater the cited discipline diversity (Table 2). |
| | (b) the interdisciplinarity of the group | The more discipline diversity in a group, the greater the cited discipline and publication diversity (Table 2) |
| | (c) There is a **negative** relationship between interdisciplinary collaboration and the diversity of international membership | There is a negative relationship between publication diversity and the diversity of countries involved (Table 2) |
| H2 | (a) There is a positive correlation between interdisciplinary collaboration in a group and number of publications. | There is a positive relationship between the number of publications and their disciplinary diversity and the diversity of the articles cited (Table 3). |
| | (b) There is an inverted-U relationship between interdisciplinary collaboration in a group and impact as measured by the median number of citations. | Median number of citations received by a group's publications is highest for intermediate values of publication and citation diversity (Table 3 and Fig 2) |
| | (c) There is a positive correlation between interdisciplinary collaboration in a group and personal satisfaction | Respondents' satisfaction with the group was <u>positively</u> related to the number of publications the group produced and the proportion of female members in the group, but <u>negatively</u> related to the diversity of the cited publications and the diversity of countries (Table 4). |
| | (d) There is a positive correlation between interdisciplinary collaboration in a group and perceived effectiveness | Perceived group effectiveness was <u>negatively</u> related to the diversity of the cited publications. It was, however, <u>positively</u> related to the disciplinary diversity of the group (Table 4) |

more depth and some additional insights into perceptions of what made the group successful or not. It must be noted that any comments were entirely voluntary. These comments have been collated and can be viewed through the Environmental Data Initiative (link to be inserted on publication).

The quantitative analysis showed that diversity of country of origin, gender and disciplinary representation can affect group process and satisfaction with group function. There were many comments about the benefits and challenges because of disciplinary or skillset differences, while gender balance was never commented on.

Hypothesis 1b posited that there was a positive relationship between interdisciplinary collaboration and the interdisciplinarity of the group, and this was supported in the quantitative

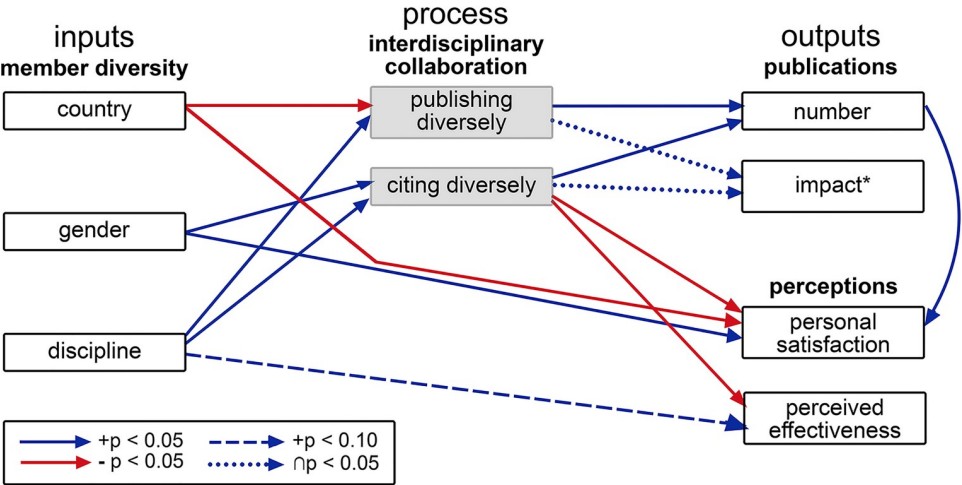

**Fig 4. Model output.** Graphical summary of the quantitative results. *specifically median citations received by each article.

analysis. In the qualitative analysis, six respondents mentioned the positive benefits of having diverse disciplines in the groups, and although there were some difficulties, respondents mentioned that these difficulties were worth dealing with. As one commented, "Slightly different objectives and different viewpoints did not always make for the most effective discussions and decisions, but I think that is part of the process and is critical to interdisciplinary groups." (C-3, R14). There was awareness of a need to deal with any challenges arising from disciplinary differences "everyone's willingness to think actively about how to improve the merger between the disciplines" (C-5, R1). Generally, this diversity was regarded as stimulating "I learnt much by simply listening to so many different scientists with different and complementary skillsets" (A-7, R1), "we brought in a diverse range of disciplines to liven up the stew." (C-2, R1). "The distance between disciplines was clearly sufficient to create some learning opportunities, but not thought to impede output" (A-7, R2). "I found the functioning of the group very effective despite people coming from different expertises" (A-3, R2).

Disciplinary differences were often spoken of interchangeably with skillset differences, "We also worked on group dynamics so different personality types and skillsets could work together and benefit from the multidisciplinarity (as opposed to becoming defensive in our corners)." (C-5, R2), "The diversity of skills represented. . .We were able to distribute the work effort in a way that made the most sense for moving the project forward." (C-4, R1). A diversity of skillsets was inherent in every group in the study, and there were reasons for separation according to skillsets but it was felt that such separation should not exclude sub-groups from overall group discussions "Everyone got along just great, and we worked hard. I would gladly work with folks again. The group somewhat naturally by necessity split into two parts; those with the technical skills for the data wrangling and statistical modelling, and those who could not. The latter group thus was able to spend a lot of time conceptualizing, a good and important task, but it left out the technical folks from that part of the project. As one of the technical folks, I would like future projects to allow a more even mix of doing both." (C-4, R5).

The value of diversity in general was, however, considered positive, ". . .I think the diversity of approaches and skills hugely contributed to the quality of the final paper. That paper was focused on how to effectively merge findings from fieldwork with the work of the modelers— that was very exciting and couldn't have happened without the diverse membership of the group." (C-5, R1). A member of another group, which comprised members from the non-research community, systems analysis, web developers, ecologists and those concerned with management and social engagement, noted that "having a diverse set of viewpoints to contribute to the work" (B-3, R1) was a factor that contributed to the group's success.

Respondents identified several additional factors beyond those we measured in the quantitative analysis. These included the diversity of background (assumed to be organisation and country of origin) of team members, career stage representation, team fidelity, including having common goals and objectives, within-group trust and respect, and the nature of leadership.

Background was highlighted occasionally as evidenced by four responses: "To put it simply, the researchers and the management agency participants initially had difficulties communicating due to different perspectives and terminology barriers" (C-3, R6). However, this diversity was regarded by another in the group as "We had somewhat different perspectives and backgrounds but mostly saw eye-to-eye, had fun and got along well." (C-3, R4). Rather interestingly, this group recognised the problem and turned it into a positive "I think the team did overcome this to some extent; indeed, one paper was largely devoted to terminology. . ." (C-3, R6). Another group observed the tension that can occur as "a lot of the collaboration depends on the diversity in the group, despite a temptation of some of the co-located members to conduct independent discussions and work without communicating with others." (B-7, R5). This

tendency for co-located members (especially if a common language was also involved) to have separate discussions excluding others in the group, was observed more than once by the authors. Ensuring this division is not disruptive is a particular role for the hosts of the groups.

The presence of early career researchers (including postdocs) was mentioned by three respondents as an important, positive, contributor to group function, "...some members of our group, young postdoctoral students who were invited by their supervisors . . . contributed a lot. Less established researchers can be a positive force in this type of working group." (A-7, R1); "The importance of the meeting and postdoc support . . . to coordinate efforts and achieve plans jointly designed during meetings" (A-1, R2), "We had a good mix of career stage: senior scientists that served as excellent advisors, we had postdocs and PhD students that have the interest and ability to work and bring new ideas, and we had motivated early career scientists. We learned a lot and some excellent publications and collaborations came out of this working group." (C-5, R3).

Sharing common goals and objectives was considered an important factor in group effectiveness, "Convergence of goals despite different backgrounds, skills and kinds of contributions" (A-5, R7); "Common interest and objectives. . ." (A-7, R2); "...consensus over the common goals." (A-8, R2); "We had clear and common goals. . ." (C-1, R5); "...dedication to tasks; belief in goals" (B-27, R6); and "The group self-selected and each member has a vested interest in the successful outcomes and outputs" (B-7, R1).

The importance of good social skills, including within-group trust and respect was raised in several ways: "Enthusiasm, nice people, no intrigue" (A-7, R4); "...mutual trust. . .and friendly atmosphere" (A-4, R1); "We listened to each other and simply got along well!" (C-1, R5); "mutual respect" (B-1, R1); "compatibility, respect, and the skills and knowledge of the members" (A-2, R4); "Tremendous individual integrity, trust in teammates, empathy in reading others' feeling, and supporting everyone on the team." (C-2, R10); and "Collaborative and friendly attitude" (B-7, R6). "It is also a great group of people who were willing to listen to others, lead, let others lead, pull their weight, be responsive, be respectful, etc. Most of these people could have big egos, but they don't. It was an outstanding professional and personal experience." (C-2, R12).

Several comments were made about the quality of leadership. If leadership was not good, things could become difficult: "...We did not have very strong group leadership, which also contributed to not getting as much done as we could have." (A-8, R4). This response incidentally was consistent with the quantitative ranking for effectiveness by group A-8 (section 4.1.3). "Effectiveness resulted from great leadership (encouraging openness to voices and ideas). . ." (C-6, R2); "Good leadership" (B-4, R1); "Leadership. Co-chairs provided an encouraging working environment where each member of the WG can thrive." (B-6, R2); "Strong leadership and vision from group PIs" (C-2, R13). As noted in the comments of R12, C-2 above, leadership need not rest with one person "Everybody's willingness to take a leadership role when needed, take a step back and letting others lead when needed, and never dismissing any idea just because it did not align with an a priori opinion" (C-6, R4).

As a final mention, however, is that these groups, as with many such groups, have a core set of enthusiasts and the face-to-face meetings were pivotal. Many of the non-targeted comments were about participation (those fence-sitters) and of the importance of the face-to-face meetings for getting work done. "A smaller subset of the group did most of the work. Many folks in the group did little or nothing" (C-5, R4); "except for a couple of fellows" (A-2, R3); and "We had a few 'doers,' that is a few key folks kept things moving. A couple of the leaders were integral in keeping a vision up front" (C-2, R3). Getting people's attention and commitment was easy within the meeting but between meetings (especially when there were a few) was harder: "most people are already overbooked for time so we had some difficulty finding people who

wanted to lead projects." (C-1, R3); "Once we went home after each meeting there was little contact and cooperation on projects. From my perspective, I think I could have made the outcome better if I had time to pursue collaborations after the face-to-face meetings. Unfortunately it was an extremely intense time in my 'real' job." (A-2, R4); "The group was very effective during the week where we were together (at least the persons that assisted to the meetings). However, much of the momentum was lost after some months" (A-1, R1); and "Working sessions went generally well. But only three to four persons were really active between working sessions." (A-3, R5). Basically "The number of meetings was super important and also helped people to have a sense of accountability. There is no substitute for being in the same room and working on projects together." (A-4, R4). This conclusion has been argued cogently by Srivastava et al. [84].

## Discussion

The goal of the work reported in this paper was to uncover the influences of group composition on the interdisciplinarity of teamwork, where the groups were attempting to find solutions to inter-disciplinary research problems. We posed a number of hypotheses about potential influences that may affect interdisciplinary practice resulting in articles, particularly articles that have impact. To examine the dynamics experienced by such groups, we used an input-mediator-output model (Fig 1) with a sample of working groups in the environmental sciences.

We found that key aspects of group composition had a largely positive effect on interdisciplinary team processes: the gender balance and the diversity of disciplines represented in the group all increased interdisciplinary collaboration (Table 5 and Fig 4), while the number of countries represented in a group offset this general trend. The number of countries is only a partial indicator of interpersonal, cultural challenges, but it is a reasonable measure which can imply a range of types of differences related to origin and background [85]. While it is a truism of research that correlation cannot prove causation, in this study the diversity of the team was set when the team was created, ruling out reverse causation, and making spurious correlation less plausible.

Our work has shown that the greater the level of interdisciplinary collaboration the greater the number of resulting publications. The impact of those publications (measured by the median level of citations received) was less straightforward, with greatest impact at moderate levels of interdisciplinary collaboration (Fig 3). It seems that being too interdisciplinary can mute impact, and too little may be 'boring'. We acknowledge that our citation data are relatively short-term (1–10 years after publication). Even though papers are usually cited most heavily shortly after publication, it is possible that more interdisciplinary papers will go on to be more cited in the longer term, as suggested in the literature [50]. The causal direction of these results might be debated, though it is not clear why teams that publish more seem to choose to do so in an interdisciplinary way, and as citations follow the process, reverse causation at least seems ruled out.

Group-member perceptions enriched our understanding of the process. Not surprisingly personal satisfaction increased with the number of publications produced and, more unexpectedly perhaps, with the proportion of women (Fig 4). Most evident from the quantitative study was the negative effect of country and diverse work practice (measured by the level of diversity of articles sourced) on personal satisfaction and perceived effectiveness of the group. There were no comments received about difficulties specifically related to nationality. The opportunity of working with people from different disciplines and backgrounds was generally considered stimulating, and respondents recognised that this discomfort did not relate to the quality of the output.

The qualitative responses highlighted that different backgrounds were often difficult at first, but adjustments could be made in the time-frame available to these groups. Four factors were identified which promoted productivity and offset any difficult collaboration barriers: (i) having early career members in the group, (ii) sharing common goals and objectives, (iii) mutual trust and respect, and (iv) the quality and team-centered nature of group leadership (as per [86]). It must be remembered that these groups worked together in a formal manner (as in having supported collaboration) for up to four years with multiple meetings during that time, and they had ample time to reflect upon and resolve their differences to produce their outcomes.

Our study examined working groups, which are scientific teams, but with differences from managerially-assembled teams employed on a research project. For instance, it is easy for a member of a voluntary working group who is dissatisfied with the team process to discontinue participation. While we believe the hypothesized relationships between demographics and process and process and outputs should not be affected by the differences, the generalizability of the model should be tested with a broader range of team types.

In summary, it can be said that the number of (relevant) disciplines brought to the table and a 'good' gender balance in a group, an interdisciplinary process can be enhanced. Open-ended feedback suggested that this would be supported by having young members in a group, and ensuring good *esprit de corps*. A strong interdisciplinary process is positively related to having more publications, with the proviso that exploring and publishing too diversely is related to reduced impact. It seems that in the case of deliberately-formed scientific working groups, with sufficient time challenges can be successfully tackled as the same group of people learn to work together.

## Conclusion

A fundamental question for our study was whether a diversity of participants in a working group does lead to diverse practices and outcomes. We think we have shown that, with some provisos and a degree of management, it does. We used an original conceptual model, namely positing the interdisciplinarity of collaboration as a moderating factor between team demographics and outcomes, and we think this has provided new insight into how groups function. The overall pattern of the findings is mostly consistent with our expectations, while the contradictory findings point to possible new perspectives on the function of research groups and a consequent management focus. We suggest that group diversity is not just a goal in itself, but is rather a support for building an interdisciplinary collaboration, the success of which has benefits for output (in moderation).

While our findings point to several factors important for working group success, they are limited by the fact that we had data from only 22 groups and 3 centers. As the survey response rates were not high, the conclusions about satisfaction and effectiveness could be more robust, and this deserves further exploration. We should also note that the groups studied were assembled by organisations that were already sensitive to some of the factors we have been discussing.

The qualitative data point to several factors important to the group participants that we were unable to capture with our quantitative data but were important to the successful function of the groups. Approaches such as the use of sociometers [87] and autoethnography [34] for example, could add greatly to the further understanding of the group dynamics that promote positive research outcomes.

Even with the limitations of our dataset, our findings reinforce the importance and apparent benefits of diversity of various types in research groups. They also indicate the need to pay

attention to the interdisciplinarity of the collaboration itself and being more deliberate about the combination of new knowledge, working to ensure that the groups bring together their diverse perspectives. The negative impacts on satisfaction and effectiveness due to diversity due to (i) participant countries and (ii) the diversity of cited publications suggests a need to help participants with the knowledge integration process to make it more enjoyable and ensure fidelity within the group.

While our study is correlational rather than causal, our findings do suggest some guidelines for the management of cross-boundary groups to successfully engage in interdisciplinary work such as:

1. ensure gender balance in a group, as this seems positively related to the interdisciplinarity of the work, the number of articles produced, and to overall group satisfaction;

2. encourage publication (in conferences or journals) throughout the group process, as group satisfaction improves with the number of publications (in contrast to the delayed publication implied in a typical Working Group Workflow);

3. be cautious if high impact articles are desired. Too much interdisciplinarity may be a distraction;

4. allow the group time and give them support to work out ways to deal with personnel differences as well managed differences can be stimulating and rewarding;

5. manage and support the discomfort that can occur when working across disciplinary boundaries, as some tension can result in novel outcomes; and

6. support the group to achieve and maintain focus on the common goal, and ensure there is mutual respect and trust between members.

As stated at the beginning of this paper, global research challenges increasingly require the formation of teams that bring together individuals with diverse skills and perspectives who can work across disciplinary, organisational and national divides. Indeed, research has become so complex that individual scientists cannot achieve meaningful results without collaborating—the so-called collaboration imperative [88]. But while interdisciplinary research is needed, will assembling a diverse team necessarily result in publications that have impact? Our results show that interdisciplinary work is, in fact, positively related to publication rates, but if those publications are themselves spread across too wide a range of disciplinary domains (perhaps reflecting the work itself), their impact may be lessened, and contrary-wise, too narrow a view will diminish impact. At the same time some other aspects of team diversity can enhance or diminish this output. Paying attention to some simple factors in the design and management of such groups can improve the likelihood of a positive research outcome and enhance the satisfaction of group members.

## Supporting information

**S1 Appendix. Demographic profiles of the groups from the contributing organisations.**
(DOCX)

**S2 Appendix. Profile of publications produced and cited by each group.**
(DOCX)

**S3 Appendix. Responses to questions about perceived effectiveness and satisfaction with the group by respondents.**
(DOCX)

## Acknowledgments

Authors would like to acknowledge DataONE for facilitating our participation in the Usability and Assessment Working Group (inter alia) and for their consistent support for this project. We would also like to acknowledge the support and contribution of the synthesis center CESAB of the French Foundation for Research on Biodiversity and Jill Baron, co-Director of the John Wesley Powell synthesis center of the USGS. We are grateful for the open and transparent work practices of all three organisations which enabled this work to be conducted. We also would like to thank the several reviewers who provided invaluable advice on drafts of the article.

## Author Contributions

**Conceptualization:** Alison Specht.

**Data curation:** Alison Specht.

**Formal analysis:** Alison Specht, Kevin Crowston.

**Investigation:** Kevin Crowston.

**Methodology:** Alison Specht, Kevin Crowston.

**Project administration:** Alison Specht.

**Writing – original draft:** Alison Specht, Kevin Crowston.

**Writing – review & editing:** Alison Specht, Kevin Crowston.

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
