## [Decision Letter · Decision Letter 0]

6 Apr 2022

PONE-D-22-04001Interdisciplinary collaboration from diverse science teams can produce significant outcomesPLOS ONE

Dear Dr. Specht,

Thank you for submitting your manuscript to PLOS ONE. After careful consideration, we feel that it has merit but does not fully meet PLOS ONE’s publication criteria as it currently stands. Therefore, we invite you to submit a revised version of the manuscript that addresses the points raised during the review process. As you can see below, the reviewers focused their reports on complementary issues. Reviewer 1, raised a number of issues regarding both quantitative and qualitative analysis (please, notice PLOS ONE's publiucation criterion #3, https://journals.plos.org/plosone/s/criteria-for-publication#loc-3). Reviewer 2 is more concerned about interdisciplinarity's conceptualisation in the article. These are relevant issues that should be addressed in the revision of the paper.

We look forward to receiving your revised manuscript.

Kind regards,

Sergi Lozano

Academic Editor

PLOS ONE

Journal Requirements:

2. In your Methods section, please provide additional information about the participant recruitment method and the demographic details of your participants. Please ensure you have provided sufficient details to replicate the analyses such as: a) the recruitment date range (month and year), b) a description of any inclusion/exclusion criteria that were applied to participant recruitment, c) a table of relevant demographic details, d) a statement as to whether your sample can be considered representative of a larger population, e) a description of how participants were recruited, and f) descriptions of where participants were recruited and where the research took place.

Reviewers' comments:

Reviewer's Responses to Questions

**Comments to the Author**

1. Is the manuscript technically sound, and do the data support the conclusions?

Reviewer #1: Partly

Reviewer #2: Partly

2. Has the statistical analysis been performed appropriately and rigorously? 

Reviewer #1: No

Reviewer #2: I Don't Know

3. Have the authors made all data underlying the findings in their manuscript fully available?

Reviewer #1: No

Reviewer #2: No

4. Is the manuscript presented in an intelligible fashion and written in standard English?

Reviewer #1: Yes

Reviewer #2: Yes

5. Review Comments to the Author

Reviewer #1: The study attempts to demonstrate how demographic diversity contributes to the interdisciplinary of collaborations and then contributes to perceptions of satisfaction and effectiveness of teams. A major contribution appears to be in its positioning as a mixed-method study. However, there are flaws with both the quantitative and qualitative aspects of the analysis. Additionally, the paper can be greatly strengthened with the inclusion of more theoretical development and analyses at the individual level. In general, the paper makes reasonable claims, but the argument needs to be strengthened and the analysis needs to be more convincing. The claims are significant, but the novelty is not particularly clear.

The paper does not effectively use literature to contextualize the research. There is little inclusion of logical mechanisms to explain the hypotheses, which reduces the potential contributions of the study. There is not much theoretical justification given to explain why these hypotheses will be present nor is there much discussion regarding the extent to which the literature provides evidence for alternative hypotheses. Specifically, there are limited references in all of Section 2.1, which diminishes the effectiveness of the hypotheses.

Sections 3 and 4 are the areas where much improvement needs to be made to help the paper. Firstly, the description of the research setting was helpful for establishing expectations and clarifying the collaborative environment. However, the variable operationalization has some opportunities for improvement. The gender balance variable does not distinguish whether there is a majority male or majority female team; either category can potentially have the same gender balance score. Another variable, such as the proportion of male or female in the group could be developed to make interpretation clearer. Currently, it seems as though information is being lost. Also, I appreciate using the median number of citations as a measure, and I would also recommend including the mean as well for a robustness check.

The explanation of the survey measures is insufficient. How many items comprise each construct? What are the items? How closely related are the items? Instead of only the mean value in the group, I would also recommend investigating the variance of responses as well. Much more information is needed to help the later interpretations of analysis given that these are group outcomes.

Once results were being presented, more analysis should be included to help support the claims. Overall, the authors mention the limitations that arise when performing statistical analyses on small sample sizes. However, I challenge the justification of the analysis on the self-reported data. There appears to be an opportunity for the researcher to conduct analysis at the individual respondent level, which would allow for the utilization of multilevel regressions where the individual responses can be nested within the group structure. Alternatively, since the group has been the level of analysis for the paper, modeling the group responses by using cluster robust standard errors would also be another approach to account for groups while analyzing the individual data. Therefore, supplementing the current analysis with analysis at the individual level would potentially improve the contributions from the paper. Additionally, providing two descriptive statistics tables would help the reader understand the group and individual data under study. To reiterate, the group level is the focus of analysis, but the lack of data makes it difficult to prove the claims, and I suggest investigating individual-level analysis to increase confidence in the analysis of self-reported data.

The Qualitative portion of the analysis was limited. It is not clear what how the inductive coding led to the emergence of themes since the presented results do not represent concepts that can be connected back to specific theoretical arguments. Additionally, there is no indication of the frequency or number of occurrences for the topics in the data. In how many teams did each of these types of findings occur? How many people included similar thoughts? Currently, the “categories” appear to be stable, and it is challenging for a reader deepen their understanding of the collaboration process.

In summary, there are positives with the paper, but there needs to be more theoretical argumentation from literature and the analysis should be supplemented to improve the evidence supporting the claims.

Reviewer #2: This is a good paper that adds to the literature on interdisciplinarity. The use of the input-process-output model to think about the role of diversity in interdisciplinary collaboration is helpful and illuminating.

There are a few limitations, though, that should be addressed in a revision before the paper is published. Principal among these is the way in which interdisciplinary collaboration is conceptualized in this study. Using the disciplinary diversity of publication venues and the disciplinary diversity of cited articles according to the journals they appear in are very blunt instruments for assessing the process of interdisciplinary collaboration, for a host of reasons. Given that interdisciplinary process is a central theme of this paper, much more time should be spent in convincing the reader that this is a valid way of conceptualizing it.

Also, I am not in a position to evaluate the statistical work in the paper, so some of my comments may be off-target because of my ignorance about this aspect of the paper.

My comments/concerns, by line(s):

l. 35. You take working groups to be a type of team. Can you say more about how a working group compares to the typical research team that is the focus of work in, say, the science of team science? These would be cohesive, interdependent groups of researchers typically with a leader or leaders and a common objective. Are you using the term ‘working group’ in the way it is used at, e.g., NCEAS? Or is there not a difference between a working group and a research team, on your way of thinking about them?

l. 87. Use of the input-process-output model is clarifying and helpful. Another discussion of interdisciplinarity where it occurs (and specifically, crossdisciplinary integration) is in O’Rourke et al. (2016): “On the nature of cross-disciplinary integration: A philosophical framework”.

ll. 95ff. In general, research teams are not composed in this way – many are pre-existing and find new problems, or are formed out of personal affinity before a problem is identified. (For a good discussion of the range, see Twyman & Contractor (2019): “Team assembly”.) Is this perhaps an answer to the question on l. 35, i.e., what are the differences between working groups and research teams? Is there a precedent for distinguishing these categories in this way?

l. 235. Focusing on the “collation and analysis of articles used and produced” seems like a very limited way of assessing the process of interdisciplinary collaboration. Can you explain and defend why you chose to do it this way?

l. 260. How was it determined whether a published paper was a paper published by a group? People may publish with others in a group and not have it be a group paper. Were these papers listed by the groups as group publications? What were the criteria here?

ll. 262ff. Given the limitations associated with CrossREF, why not use a broader and more inclusive platform, e.g., Google Scholar?

ll. 265ff. This is a potentially suggestive measure of the interdisciplinarity of an article, but why take it to be a measure of the interdisciplinarity of a collaboration? Surely an article could have a diversity of disciplines represented in its References section, but where that fact does not correspond to interdisciplinary process in the collaboration. For example, the co-authors could be responsible for introducing literature from their disciplines without there being any interdisciplinary collaboration involved. The same is true for the output measure – a group might decide to publish in a range of journals so that teammates can receive credit in their home departments by publishing in a journal the department recognizes, without that entailing any actual disciplinary integration in the production of those papers. Since your decision to center interdisciplinary process is really the crux of this paper, you need to develop and defend this choice in more detail, IMO. And you should also address this as a limitation in the discussion.

l. 295. Did you ask any questions about interdisciplinary process in the survey? Did you receive any open-ended responses that had a bearing on how you understand interdisciplinary process?

l. 309. What was the overall response rate to the survey?

ll. 465ff. Why did you choose the A groups as the control for the group factor? (And I guess I’m not sure what you mean by “group factor” here?)

ll. 519ff. For readability, I would encourage you to use a colon or comma to separate your quote introductions from the quotes themselves, e.g., ll. 535-7 where I was tempted by the lack of punctuation to read that as one long (and very awkward) sentence.

l. 636. Were there really no changes in the working groups over time that affected their diversity? If there were, how does that affect what you’re saying here? Does it have any impact on your analyses in this paper?

l. 655. Maybe supply a couple of examples of these suggestions?

l. 670. Might it not take more time for a working group that publishes widely to have impact? How can you distinguish between differences in impact vs. differences in rate of impact?

6. PLOS authors have the option to publish the peer review history of their article (what does this mean?). If published, this will include your full peer review and any attached files.

Reviewer #1: No

Reviewer #2: No

---

## [Author Response · Author response to Decision Letter 0]

13 Sep 2022

I have uploaded a comprehensive (I hope) response to reviewers and editor's document. It is formatted as a table.

Response to reviewers

editor’s comments 

We believe we did meet this and have further checked our formatting, moving Appendices to Supplementary material, among other things. Further specifics will be complied with if we have overlooked them.

2. In your Methods section, please provide additional information about the participant recruitment method and the demographic details of your participants. Please ensure you have provided sufficient details to replicate the analyses such as: a) the recruitment date range (month and year), b) a description of any inclusion/exclusion criteria that were applied to participant recruitment, c) a table of relevant demographic details, d) a statement as to whether your sample can be considered representative of a larger population, e) a description of how participants were recruited, and f) descriptions of where participants were recruited and where the research took place.

 We have added the requested information in the methods and results section. The first Appendix (S1) has relevant demographic details. 

3. In your Data Availability statement, you have not specified where the minimal data set underlying the results described in your manuscript can be found. PLOS defines a study's minimal data set as the underlying data used to reach the conclusions drawn in the manuscript and any additional data required to replicate the reported study findings in their entirety. All PLOS journals require that the minimal data set be made fully available. For more information about our data policy, please see http://journals.plos.org/plosone/s/data-availability

We have made a statement in the body of the paper about the ethics conditions related to the data. This restricts us with respect to the full identity of the respondents, their journal publications and the centres with which they were associated.

We have uploaded the Appendices as Supporting Information (Appendices), and the data as mentioned.

4. Please note that in order to use the direct billing option the corresponding author must be affiliated with the chosen institute. Please either amend your manuscript to change the affiliation or corresponding author, or email us at plosone@plos.org with a request to remove this option. The corresponding author has been changed and is now Prof. Kevin Crowston of Syracuse University.

5. Please include your full ethics statement in the ‘Methods’ section of your manuscript file. In your statement, please include the full name of the IRB or ethics committee who approved or waived your study, as well as whether or not you obtained informed written or verbal consent. If consent was waived for your study, please include this information in your statement as well. We have included a section in the Methods as an introduction to the IRB conditions for the whole project. 

reviewer comments

Reviewer 1

1.The study attempts to demonstrate how demographic diversity contributes to the interdisciplinary of collaborations and then contributes to perceptions of satisfaction and effectiveness of teams. 

A major contribution appears to be in its positioning as a mixed-method study. However, there are flaws with both the quantitative and qualitative aspects of the analysis. Additionally, the paper can be greatly strengthened with the inclusion of more theoretical development and analyses at the individual level. In general, the paper makes reasonable claims, but the argument needs to be strengthened and the analysis needs to be more convincing. The claims are significant, but the novelty is not particularly clear. Yes, one aspect of the paper is to assess demographic diversity and its effect on interdisciplinary processes. Assuming disciplinary diversity of group members is a demographic metric, then yes.

We address the detailed suggestions below. 

2.The paper does not effectively use literature to contextualize the research. There is little inclusion of logical mechanisms to explain the hypotheses, which reduces the potential contributions of the study. There is not much theoretical justification given to explain why these hypotheses will be present nor is there much discussion regarding the extent to which the literature provides evidence for alternative hypotheses. Specifically, there are limited references in all of Section 2.1, which diminishes the effectiveness of the hypotheses. We use the input-process-output model to structure our selection of factors and their relationships. We have added to the discussion of the hypotheses and do indicate the mechanism by which the chosen group composition factors are believed to affect the group process and for why the group process should affect the outcomes. Section 2.1 now includes approximately 30 citations to past studies supporting the hypotheses. There are of course additional studies that suggest alternative explanations, but we did not think it useful to include hypotheses that are beyond our focus on interdisciplinarity or for which we do not have data. 

3. Sections 3 and 4 are the areas where much improvement needs to be made to help the paper. Firstly, the description of the research setting was helpful for establishing expectations and clarifying the collaborative environment. 

However, the variable operationalization has some opportunities for improvement. 

The gender balance variable does not distinguish whether there is a majority male or majority female team; either category can potentially have the same gender balance score. Another variable, such as the proportion of male or female in the group could be developed to make interpretation clearer. Currently, it seems as though information is being lost. Also, I appreciate using the median number of citations as a measure, and I would also recommend including the mean as well for a robustness check. As suggested, we now analyze the proportion of females in the group, which is a measure that has been used in other studies of group dynamics. The main results are unchanged. 

As requested, we ran a regression to predict average citations instead of median. As average is a non-robust measure, the data have more outliers, which required a robust regression. The data were not over-dispersed, so we used a Poisson regression. In this regression, age was still a significant predictor (older groups are cited more on average, as would be expected) and there were significant differences among the centres in average citation. Publication and citation diversity still showed a curvilinear relationship (more citations for moderate levels of diversity). Since the results are qualitatively similar but require a more complex test, we feel it is preferable to report the results for median citations. 

4. The explanation of the survey measures is insufficient. How many items comprise each construct? What are the items? How closely related are the items? 

Instead of only the mean value in the group, I would also recommend investigating the variance of responses as well. Much more information is needed to help the later interpretations of analysis given that these are group outcomes. We only had a few survey measures to include in the analysis. We have added text to section 3.2.3 to explain the measures used. We had originally conceptualized perceived effectiveness and satisfaction as being sub-items of a measure of group performance, along with output, but when we discovered they were not well correlated we decided to treat them separately. However, we had only included a single item for each. This is because the questions were part of a longer survey (many of the survey questions were not on the topic of interdisciplinarity so were not analysed in this paper). Because the survey was quite long we considered that having too many items for scales would be problematic for return rate. 

We note that Appendix 3 giving the variances (standard errors) of the measures was omitted from the paper, and it has been re-inserted.

5. Once results were being presented, more analysis should be included to help support the claims. Overall, the authors mention the limitations that arise when performing statistical analyses on small sample sizes. However, I challenge the justification of the analysis on the self-reported data. There appears to be an opportunity for the researcher to conduct analysis at the individual respondent level, which would allow for the utilization of multilevel regressions where the individual responses can be nested within the group structure. 

Alternatively, since the group has been the level of analysis for the paper, modeling the group responses by using cluster robust standard errors would also be another approach to account for groups while analyzing the individual data. Therefore, supplementing the current analysis with analysis at the individual level would potentially improve the contributions from the paper. Additionally, providing two descriptive statistics tables would help the reader understand the group and individual data under study. 

To reiterate, the group level is the focus of analysis, but the lack of data makes it difficult to prove the claims, and I suggest investigating individual-level analysis to increase confidence in the analysis of self-reported data. We re-examined the data we had on individuals with an eye towards carrying out an individual-level analysis. However, we note that the constructs in our model are all defined at the group level and do not seem meaningful at the individual level. For instance, it is not clear what it would mean for an individual to have an interdisciplinary group or process by him or herself, nor how to measure it in a way that does not end up as a group measure. We therefore have kept the focus in the paper at the group level.

6. The Qualitative portion of the analysis was limited. It is not clear what how the inductive coding led to the emergence of themes since the presented results do not represent concepts that can be connected back to specific theoretical arguments. 

Additionally, there is no indication of the frequency or number of occurrences for the topics in the data. In how many teams did each of these types of findings occur? How many people included similar thoughts? Currently, the “categories” appear to be stable, and it is challenging for a reader [to] deepen their understanding of the collaboration process. We have revised the presentation of the qualitative results to make it clearer where the comments connect to concepts in our original theoretical model or to related concepts (e.g., other kinds of demographic diversity beyond what we measured). In addition, the comments revealed additional concerns that went beyond our model, e.g., the importance of leadership, which we report for completeness even though they are not related to diversity or interdisciplinarity. 

We do not believe that turning qualitative data into quantitative numbers is appropriate. Since respondents were choosing what topics to mention, there’s no reason to believe that a theme being mentioned more often translates directly to it being more important, e.g., the lack of mention of the impact of gender, despite prior research and our results showing its impact. However, we chose quotations uniformly across the themes, meaning that there is a relation between how often a theme appeared in the data and how often we include a quotation for the theme. 

7. In summary, there are positives with the paper, but there needs to be more theoretical argumentation from literature and the analysis should be supplemented to improve the evidence supporting the claims. We hope that the added theoretical argumentation and analysis addresses your concerns. 

reviewer 2

1.This is a good paper that adds to the literature on interdisciplinarity. The use of the input-process-output model to think about the role of diversity in interdisciplinary collaboration is helpful and illuminating.

There are a few limitations, though, that should be addressed in a revision before the paper is published. Principal among these is the way in which interdisciplinary collaboration is conceptualized in this study. Using the disciplinary diversity of publication venues and the disciplinary diversity of cited articles according to the journals they appear in are very blunt instruments for assessing the process of interdisciplinary collaboration, for a host of reasons. Given that interdisciplinary process is a central theme of this paper, much more time should be spent in convincing the reader that this is a valid way of conceptualizing it. Thank you for your interest. 

We have added text at the start of section 3.2.2 about the rationale for using the bibliometric method. 

We have added a detailed analysis in 4.1.2 of two (three) groups that score high or low in the bibliographic diversity scale. We have added a comment about the per article citation rate.

2. Also, I am not in a position to evaluate the statistical work in the paper, so some of my comments may be off-target because of my ignorance about this aspect of the paper. We have kept your disclaimer in mind in responding to your suggestions. 

l. 35. You take working groups to be a type of team. Can you say more about how a working group compares to the typical research team that is the focus of work in, say, the science of team science? These would be cohesive, interdependent groups of researchers typically with a leader or leaders and a common objective. Are you using the term ‘working group’ in the way it is used at, e.g., NCEAS? Or is there not a difference between a working group and a research team, on your way of thinking about them? We have revised the Introduction to address the characteristics of the working groups as scientific teams, in paragraph one, line 3 onwards. 

We have added some text and a figure in section 3.1 Research Setting, to better illustrate the scenario and explain, hopefully better, the voluntary and part-time nature of the collaboration being studied.

l. 87. Use of the input-process-output model is clarifying and helpful. Another discussion of interdisciplinarity where it occurs (and specifically, cross disciplinary integration) is in O’Rourke et al. (2016): “On the nature of cross-disciplinary integration: A philosophical framework”. Thank you for highlighting this reference for us.

We have added it as a reference with respect to grand challenges. But this thoughtful paper has been added to by Bammer et al., 2020, which has also been added.

ll. 95ff. In general, research teams are not composed in this way – many are pre-existing and find new problems, or are formed out of personal affinity before a problem is identified. (For a good discussion of the range, see Twyman & Contractor (2019): “Team assembly”.) Is this perhaps an answer to the question on l. 35, i.e., what are the differences between working groups and research teams? Is there a precedent for distinguishing these categories in this way? This is a most helpful point. 

There is a quote right at the start of Twyman and Contractor’s section “An outside authority is responsible for the performance of a staffed team, while self-assembled teams are responsible for their own success.” This is not quite the same for these synthesis groups, as the synthesis centre/dataone are keen to see outcome, but it is getting there. We have added a comment. 

l. 235. Focusing on the “collation and analysis of articles used and produced” seems like a very limited way of assessing the process of interdisciplinary collaboration. Can you explain and defend why you chose to do it this way? We have added text arguing for this method at the start of 3.2.2.

l. 260. How was it determined whether a published paper was a paper published by a group? People may publish with others in a group and not have it be a group paper. Were these papers listed by the groups as group publications? What were the criteria here? Good question.

(a) the groups and the centre/DataONE management team kept a close eye on the publications (e.g., providing lists of papers by group), and (b) the teams may have occasionally brought other authors into a publication, but the initiative was the groups and the majority of authors were from the group.

ll. 262ff. Given the limitations associated with CrossREF, why not use a broader and more inclusive platform, e.g., Google Scholar? We could have used several sources and created a system of amalgamating them. However, it is, in our experience, unlikely to change the pattern of the results. We have queried several publishing house colleagues (e.g., DataCite) and CrossREF is accepted as pretty comprehensive for English-language articles. In other words, as long as the source of citation data is not systematically biased, then the limitations should not change the results. We have added an explanation to this paragraph.

ll. 265ff. This is a potentially suggestive measure of the interdisciplinarity of an article, but why take it to be a measure of the interdisciplinarity of a collaboration? Surely an article could have a diversity of disciplines represented in its References section, but where that fact does not correspond to interdisciplinary process in the collaboration. For example, the co-authors could be responsible for introducing literature from their disciplines without there being any interdisciplinary collaboration involved. 

The same is true for the output measure – a group might decide to publish in a range of journals so that teammates can receive credit in their home departments by publishing in a journal the department recognizes, without that entailing any actual disciplinary integration in the production of those papers. Since your decision to center interdisciplinary process is really the crux of this paper, you need to develop and defend this choice in more detail, IMO. And you should also address this as a limitation in the discussion. We explain our choice of measure in the paper as follows.

“As we were unable to follow each group individually over the several years they were collaborating, we sought a measure that could be applied retrospectively. Interdisciplinarity can be assessed in many ways, of which bibliometric measures are the most developed and have valuable antecedent analyses (e.g. Wagner et al. [59], Leyesdorf and Rafols, 2011, Rafols et al., 2012, Marx and Bornemann, 2016, Leydesdorf et al., 2018).

To assess the interdisciplinarity of the working groups’ collaborations, we analysed the literature on which the group drew to produce their published articles, the ‘inspiration’ measure of [7] and used by several authors in whole or in part as a measure of interdisciplinarity [29], and [60], Huang et al., 2022, Shu et al, 2022).”

We think it unlikely that authors could draw on literature from different domains in a paper without offering any synthesis or integration, that is, to be multi-disciplinary in a single paper. The scenario suggested where members of the group publish in their home disciplines without any interdisciplinary integration would yield a high score for publication diversity but a low score for citation diversity, since the individual disciplinary papers would not need (or be able to) to cite outside the discipline, so that effect would be visible in our analyses.

l. 295. Did you ask any questions about interdisciplinary process in the survey? Did you receive any open-ended responses that had a bearing on how you understand interdisciplinary process? We asked about factors that helped or hindered effectiveness, satisfaction and general outcomes. The comments about disciplinary differences are listed in section 4.3

l. 309. What was the overall response rate to the survey? The response rate has been inserted.

ll. 465ff. Why did you choose the A groups as the control for the group factor? (And I guess I’m not sure what you mean by “group factor” here?) When running a regression using factor variables (those having discrete unordered levels), the typical approach is to represent the levels of the factors as dummy variables. One level of the factor is picked as the base and the regression identifies how the other levels differ from that base. The default in R is to use the first factor level as the base. Specifically, we are comparing data for working groups from three centres, so the variable representing the centre is a factor variable with 3 levels. The first level is arbitrarily selected as base value (i.e., the A groups), and the regression includes two dummy variables representing membership in the B or the C centre respectively. The regression weights for those dummy variables indicates how much on average a group in centre B or C differ from those in centre A, controlling for the other variables. 

ll. 519ff. For readability, I would encourage you to use a colon or comma to separate your quote introductions from the quotes themselves, e.g., ll. 535-7 where I was tempted by the lack of punctuation to read that as one long (and very awkward) sentence. We have inserted semicolons regularly, and ‘ands’ as linking.

l. 636. Were there really no changes in the working groups over time that affected their diversity? If there were, how does that affect what you’re saying here? Does it have any impact on your analyses in this paper? There was little change in the group composition at the times of the surveys (2014 and 2018) in relation to the start of the groups. This could only be ascertained, however, from records of meeting attendance and some unsolicited comments. Meeting attendance data was not available consistently for all groups, and for the data that were available, fidelity to the group as evidenced by meeting attendance was between 60% and 80% over 2-10 meetings. Fidelity was always greater than 80% for groups which had only two meetings. Some of this text has been entered in the paper.

Feedback has been included in section 4.3 that mentions the classical ‘committed’ and ‘less committed’ members of groups.

l. 655. Maybe supply a couple of examples of these suggestions? These are already mixed in the qualitative section (4.3), and were mainly exhortations to exercise patience. This sentence was removed from the main body of the paper.

l. 670. Might it not take more time for a working group that publishes widely to have impact? How can you distinguish between differences in impact vs. differences in rate of impact? Some of our working groups were quite old (e.g., created in 2010) while others were young, so we believe the data are sufficient to capture impact. We control for age of the working groups to enable comparisons, as we expect groups to be more productive (have more papers out) and have more citations with age. 

Analyzing the rate of impact retrospectively would be quite challenging as it would require identifying the timing of each individual citation. As we are already controlling for age, this additional analysis was not pursued.

---

## [Decision Letter · Decision Letter 1]

18 Oct 2022

PONE-D-22-04001R1Interdisciplinary collaboration from diverse science teams can produce significant outcomesPLOS ONE

Dear Dr. Crowston,

Thank you for submitting your manuscript to PLOS ONE. After careful consideration, we feel that it has merit but does not fully meet PLOS ONE’s publication criteria as it currently stands. Therefore, we invite you to submit a revised version of the manuscript that addresses the points raised during the review process.

As you can see below, both reviewers are satisfied with the new version of the manuscript. I am fine with that. However, Reviewer 2 pointed out to some minor issues that should be fixed before proceeding towards publication.

Moreover, Reviewer 2 also disagrees with some of the statements made in the text about interdisciplinarity and how to measure it. I would like to ask you to answer his comments (considering the relevance of the issue) and, eventually, referring to this debate in the manuscript (if you find it interesting enough).

We look forward to receiving your revised manuscript.

Kind regards,

Sergi Lozano

Academic Editor

PLOS ONE

Journal Requirements:

Reviewers' comments:

Reviewer's Responses to Questions

**Comments to the Author**

1. If the authors have adequately addressed your comments raised in a previous round of review and you feel that this manuscript is now acceptable for publication, you may indicate that here to bypass the “Comments to the Author” section, enter your conflict of interest statement in the “Confidential to Editor” section, and submit your "Accept" recommendation.

Reviewer #1: All comments have been addressed

Reviewer #2: (No Response)

2. Is the manuscript technically sound, and do the data support the conclusions?

Reviewer #1: (No Response)

Reviewer #2: Yes

3. Has the statistical analysis been performed appropriately and rigorously? 

Reviewer #1: (No Response)

Reviewer #2: I Don't Know

4. Have the authors made all data underlying the findings in their manuscript fully available?

Reviewer #1: (No Response)

Reviewer #2: Yes

5. Is the manuscript presented in an intelligible fashion and written in standard English?

Reviewer #1: (No Response)

Reviewer #2: Yes

6. Review Comments to the Author

Reviewer #1: I greatly appreciate the authors' decisions in how they chose to address my comments, questions, and concerns. Overall, the paper is more effective in its goal and the main document does not have the same issues. I have no issues with this paper being accepted due to the enhanced literature review and additional analyses that were conducted at my recommendation.

Reviewer #2: This is a good paper, and one worth publishing. I appreciate the work the authors did to improve the paper in light of the previous round of comments. I think the results are especially interesting and helpful. I suspect I will cite this paper in the future.

FWIW, I disagree with both parts of this reply to my previous comment on ll. 265ff:

"We think it unlikely that authors could draw on literature from different domains in a paper without offering any synthesis or integration, that is, to be multi-disciplinary in a single paper. The scenario suggested where members of the group publish in their home disciplines without any interdisciplinary integration would yield a high score for publication diversity but a low score for citation diversity, since the individual disciplinary papers would not need (or be able to) to cite outside the discipline, so that effect would be visible in our analyses."

I have been in more than one crossdisciplinary collaboration that generated papers which would appear interdisciplinary on the authors assumptions but were decidedly multidisciplinary in their construction: different authors were given responsibility for different sections, and they brought their references along with them. And also in my experience, if a group publishes papers in the home disciplines of their members while keeping those members as co-authors (e.g., if you have the same set of co-authors but different first authors), the publications will often share a multidisciplinary set of references in common. (Perhaps there are some disciplines whose journals discourage citation outside the discipline, but I have not encountered that.)

That said, the authors are correct that the bibliometric methods they use as proxies for interdisciplinary collaboration are standard in the literature. I think they are weak measures of interdisciplinary collaboration, and as such represent a substantial limitation of this paper, but I don’t think it is fair to hold that against the authors since they are cleaving to standard practice here.

One general comment: I find the numbering helpful in keeping straight how you are nesting the sections and subsections. Perhaps this is inconsistent with PLOS One style, though? At this point, I have difficulty keeping track of which are the main sections, which are the subsections, and which are the sub-subsections.

I do have a few small comments/suggestions, none of which are deal-breakers. By line number:

l. 85 Reference [5] is repeated in here as reference [29].

ll. 112-3 This still seems too fast to me. You can have a diverse team with very little integration – e.g., a multidisciplinary team that has one person (say, the PI) who is especially committed to broad citation. “[T]he diversity of authors and the references they cite” give you very little information about whether any of the processes mentioned in the quote from [33] actually took place in the team. IMO, of course. (Not that you are in a position to do better than this, given that you weren’t part of these groups – see the comment above.) It would make me a bit happier to see the authors note the gap between the processes mentioned in the quote and the proxy variables mentioned in the last sentence.

l. 185 Should be ‘have’, not ‘has’.

ll. 205-8 I have difficulty parsing the clause that includes two instances of "accepted by" – is there something missing between "university" and the second "accepted"?

l. 372 Space missing between ‘members’ and ‘who’.

ll. 479-81 On what is this claim based?

l. 543 If you make reference to the section numbering, you should number your sections.

ll. 669-70 The two "but" clauses in this sentence makes it difficult to interpret. Rewrite?

ll. 682-3 Not sure how to read this last clause. Is it supposed to be another example of positive consideration of the value of diversity in general? Maybe set its context up a bit more?

ll. 789-90 Would it be fair to say that your results only show muted short-term impact?

ll. 800-3 Perhaps this is not surprising, given that you only draw from groups that were productive and productivity is positively correlated with satisfaction (ll. 481-3).

ll. 868-70 Not sure what you mean by "staged output"? What findings suggest this? I am not sure I can locate what in your Results section supports this guideline.

l. 874 You don't need the commas in (4) – they make it hard to parse.

7. PLOS authors have the option to publish the peer review history of their article (what does this mean?). If published, this will include your full peer review and any attached files.

Reviewer #1: No

Reviewer #2: **Yes: **Michael O'Rourke

---

## [Author Response · Author response to Decision Letter 1]

6 Nov 2022

Please see attached response to reviewers document.

---

## [Editor Report · Decision Letter 2]

9 Nov 2022

Interdisciplinary collaboration from diverse science teams can produce significant outcomes

PONE-D-22-04001R2

Dear Dr. Crowston,

We’re pleased to inform you that your manuscript has been judged scientifically suitable for publication and will be formally accepted for publication once it meets all outstanding technical requirements.

Kind regards,

Sergi Lozano

Academic Editor

PLOS ONE
---

## [Editor Report · Acceptance letter]

17 Nov 2022

PONE-D-22-04001R2 

Interdisciplinary collaboration from diverse science teams can produce significant outcomes 

Dear Dr. Crowston:

I'm pleased to inform you that your manuscript has been deemed suitable for publication in PLOS ONE. Congratulations! Your manuscript is now with our production department. 

Kind regards, 

on behalf of

Dr. Sergi Lozano 

Academic Editor

PLOS ONE